# Design and High-Resolution Analysis of an Efficient Periodic Split-and-Recombination Microfluidic Mixer

**DOI:** 10.3390/mi13101720

**Published:** 2022-10-12

**Authors:** Xiannian Zhang, Zhenwei Qian, Mengcheng Jiang, Wentao Li, Yanyi Huang, Yongfan Men

**Affiliations:** 1School of Basic Medical Sciences, Advanced Innovation Center for Human Brain Protection, Capital Medical University, Beijing 100069, China; 2Biomedical Pioneering Innovation Center (BIOPIC), Peking-Tsinghua Center for Life Sciences, Beijing Advanced Innovation Center for Genomics (ICG), College of Chemistry and Molecular Engineering, Peking University, Beijing 100871, China; 3Research Center for Biomedical Optics and Molecular Imaging, Institute of Biomedical and Health Engineering, Shenzhen Institutes of Advanced Technology, Chinese Academy of Sciences, Shenzhen 518055, China

**Keywords:** microfluidic mixer, microfabrication, lab-on-chip, uTAS

## Abstract

We developed a highly efficient passive mixing device based on a split-and-recombine (SAR) configuration. This micromixer was constructed by simply bonding two identical microfluidic periodical open-trench patterns face to face. The structure parameters of periodical units were optimized through numerical simulation to facilitate the mixing efficiency. Despite the simplicity in design and fabrication, it provided rapid mixing performance in both experiment and simulation conditions. To better illustrate the mixing mechanism, we developed a novel scheme to achieve high-resolution confocal imaging of serial channel cross-sections to accurately characterize the mixing details and performance after each SAR cycle. Using fluorescent IgG as an indicator, nearly complete mixing was achieved using only four SAR cycles in an aqueous solution within a device’s length of less than 10 mm for fluids with a Péclet number up to 8.7 × 10^4^. Trajectory analysis revealed that each SAR cycle transforms the input fluids using three synergetic effects: rotation, combination, and stretching to increase the interfaces exponentially. Furthermore, we identified that the pressure gradients in the parallel plane of the curved channel induced vertical convection, which is believed to be the driving force underlying these effects to accelerate the mixing process.

## 1. Introduction

Miniaturized and automated microfluidic devices have been widely used in various analytical platforms that need to handle a small volume of liquid [1,2]. In many of these applications, the size of the device is compact, and typically in the format of a ‘chip’ [3]. Such chip-based devices have greatly facilitated experimental operations in research laboratories and in clinical diagnostics especially when a point-of-care test is needed [4,5]. One of the main operations of liquid-based analysis is mixing, which is always challenging in microfluidic devices [6,7]. The Reynolds number (Re) in micro-channels, with a typical dimension from tens to hundreds of micrometers, is very small and defines the laminar-dominant flow in these microfluidic devices [8]. Therefore, fluid mixing mainly relies on passive diffusion, which is slow for large biomolecules and particles.

Many efforts have been made to improve micro-scale mixing through elaborate device and channel designs [9,10]. Although active mixers, which count on external forces to perturb liquid flow, are commonly integrated into compact devices, the design and fabrication processes are typically much more complex than constructing passive mixers [11,12]. Some specifically designed channel structures are able to increase the area of contacting interface between fluids to enhance molecular diffusion [13,14]. Chaotic advection and multi-lamination are the two major approaches to creating such enhancement in various passive mixers [15,16]. Chaotic advection can be introduced by specific channel designs such as adding grooves to the channel [9] or using twisted [17] or serpentine channels [18,19]. Chaotic advection devices are easy to design and fabricate but rely on relatively larger Re, which limits their usage in applications with low flow speed or smaller scale [20,21]. On the other hand, multi-lamination flow is mainly formed by manipulating and folding the laminar flow within the microchannels [22]. Such devices, typically creating multiple interfaces between fluids, are more efficient at low flow speeds under which the diffusion becomes more effective [23]. In fact, chaotic advection and multi-lamination effects can be combined through properly designed micro-channel patterns, such as the split-and-recombine (SAR) structure [19,24,25,26]. Although many SAR-based micromixers have been reported, most devices are laborious to fabricate due to their adoption of customized molds using 3D printing or using complex channel patterns [23,27,28]. In addition, the detailed mixing mechanisms are not comprehensively addressed using numerical simulation and high-resolution imaging analysis [18,19,25,29].

In this work, we demonstrated a novel schema of micromixers following the SAR strategy. In the design, the three-dimensional fluidic splitting and recombination are realized through symmetrically patterned channels, which can be simply fabricated through face-to-face bonding of two identical open-channel slabs cast from the same lithography master mold. To acquire high-resolution confocal imaging profiling of mixing results, we introduced optical transparent edges to the PDMS slabs of mixers. This SAR micromixer is extremely efficient, with a mixing ratio of 91% for goat anti-rabbit IgG in an aqueous solution after four SAR cycles within a length of 10 mm, measured with a newly developed high-resolution imaging method and verified by finite-element numerical simulations. Further analysis has been carried out to help understand the mechanism of efficient mixing and to optimize the channel design to achieve improved performance. This highly efficient SAR passive micromixer design can be well integrated into most compact microfluidic devices.

## 2. Materials and Methods

### 2.1. The Design of Micromixer

The schematic structure of the three-dimensional SAR micromixer (Figure 1a–d) is formed from two layers of open-channel microscale patterns. The structure is periodic, and the cross-section of the structure is centrosymmetric. The general idea of efficient mixing is to simultaneously generate advection and multiple diffusion interfaces. The two fluidic streams, separately injected into the device through separate inlets, are brought to mix with each other at the beginning of the first periodical unit. In each unit, the two input streams are first aligned and then rotationally split into two arms, one on the top layer and the other on the bottom layer (Figure 1b). At the mid-positions of each period, both streams vertically switch to the opposite layers and then re-merge at the end of the unit. In each such SAR unit, the fluid at the diagonal corners will be brought together, creating a stretched diffusion interface in between. The design of the micromixer contains several key geometrical parameters of the channels, including the length of the junction region (L), height (H), and curvature (R and θ).

### 2.2. The Fabrication of PDMS SAR Micromixer

The fabrication procedure of this SAR micromixer device is simple in both conception and operation. Two identical slabs with open trenches (Figure 1e) were bonded face to face, to form three-dimensional microchannels with inversion symmetry of two slabs. Although such a fabrication approach does not specifically require certain materials in this work, we chose PDMS to demonstrate the whole process due to the easiness of the soft lithography. Two PDMS slabs can be cast from the same master mold, which can be made from conventional photolithography using SU-8 negative photoresist (SU-8 2050, MicroChem Corporation, Newton, MA, USA) on silicon wafers. Channel height is controlled by the speed of SU-8 spin-coating. After two partially cured PDMS slabs (Sylgard 184, Dow Corning, 10:1, cured at 80 °C for 30 min) were peeled from the mold, inlet holes were punched on one of the slabs and connected with tygon tubes. Then the two slabs were carefully aligned under a stereoscope and bonded through complete curing at 80 °C overnight. To improve the precision and accuracy of both device fabrication and confocal imaging, we scaled the optimized simulation design by a scale factor of two. The channel width is 200 µm, the single layer channel height is 60 µm (SU-8 2050, 2500 rpm, 1 min), and the R is 1200 µm with the curvature unchanged. The length of the four mixing units of the mixer device is about 8.75 mm.

### 2.3. Numerical Simulation

For a better understanding of the micromixer and to pursue the improvement of mixing efficiency, the flow and diffusion between the two input solutions were numerically analyzed to quantitatively assess the fluid status in the channel. A simulation was performed with a commercially available computational fluid dynamics (CFD) package (FLUENT 6.2). The mesh process was carried out with GAMBIT 2.2, with cubic grids and tetrahedral meshes. More than 2,000,000 cells were applied for each SAR cycle in the mesh process. The inlet boundary condition was set to have a constant velocity for both fluids, and the outlet was set to have a constant pressure condition. Glycerin solution (*v*/*v* 65%) at 25 °C was used as the working fluid, and the inlet velocity was set to be 20 mm/s (0.12 µL/s). The results were visualized in TECPLOT 360 (version 15.1, Bellevue, WA, USA). Volume fraction distributions of the fluids in the outlet were obtained through simulation. To evaluate the mixing performance, the simulated mixing ratio was calculated with the following equation as in previous work [22]:(1)$$R=\left(1-\sqrt{\frac{1}{N}\overset{N}{\underset{i=1}{\sum }}{\left(I_{i}-I_{i}^{perf.mix}\right)}^{2}/\frac{1}{N}\overset{N}{\underset{i=1}{\sum }}{\left(I_{i}^{0}-I_{i}^{perf.mix}\right)}^{2}}\right)\times 100\%$$
in which *N*, *I_i_*, $I_{i}^{0}$, and $I_{i}^{perf.mix}$ represent the total number of pixels, the fluorescence intensity at pixel *i*, the fluorescence intensity at pixel *i* without mixing, and the fluorescence intensity of the completely mixed solution at pixel *i*, respectively.

### 2.4. Experimental Setup

To validate the performance of the mixer, two fluids were injected into the inlets, and the fluid distribution of the cross-section (the x-z plane) of the outlet was imaged with a confocal microscope directly and compared to the simulated result. In the previous reports, the cross-section images were typically reconstructed from three-dimensional confocal data collected from the direction perpendicular to the flow direction [22]. The resolution of the z-direction is hence significantly lower than that of the x-direction. In addition, z-sectioning also takes longer than imaging a single frame. To best exploit the high resolution of two-dimensional confocal images we decided to directly image the x-z cross-sections by observing the flow directions. However, since the working distance of the microscope objective is limited (2.2 mm for Leica objective HCX PL APO 10×/0.40 CS), we then developed a new method to acquire the confocal images of cross-sections at the specific positions of the mixer. We designed a few specific mixer devices consisting of a different number of SAR units and acquired the cross-section images at the end joint of the last unit. To create a clear end facet for image taking, we cured a thin layer of PDMS onto the side surface that is close to the imaging plane, using the method we developed in our previous work (Figure 2a) [30]. Briefly, the cured PDMS device was cut with a blade to create a side surface that is close to the selected cross-section with a distance of no more than 1 mm, which is within the working distance of the objective lens. We then span a thin layer of uncured PDMS on a silicon wafer and stamped the chip with the side surface toward it. After being cured at 80 °C for 30 min, the chip was peeled off and the mirror-flat side surface was formed with great transparency. Two syringe pumps were connected to inputs using tygon tubing and the flow rate was controlled by injection rate, and the highest flow speed we tested was 20 mm/s (0.48 µL/s) in channels to prevent the PDMS chip from delamination. One input was dyed with red-fluorophore-labeled goat anti-rabbit IgG (0.2 mg/mL, ~160 kDa, diffusion coefficient: 4.6 × 10^−11^ m^2^/s) as the tracer for fluorescence confocal imaging.

## 3. Results

### 3.1. The Design and Optimization of the SAR Micromixer Structure

Although the general sketch of the design is simple, the geometrical parameters of the channels, including the length, height, and curvature, are critical factors that will determine the performance of mixing. To identify the optimized design, we calculated the mixing ratio of two fluids, illustrated as the color blue and red in Figure 3a, through simulation of mixers with different geometries. The spatial distribution of two fluids in certain representative cross-sections are ideal illustrations for understanding the mixing process.

There are a few practical considerations of channel sizes. The total volume should be kept small to reduce the dead volume. However, the size of the channel should not be too small otherwise the high-pressure resistance will cause many difficulties in device control, such as the easiness of leakage. The typical dimension of the width of the channels is between 10 µm to 1 mm, thus we adopted a moderate and prevalent channel width of 100 μm. We have found that at such a scale, the radius of curvature (R) does not significantly affect the mixing performance (Figure 3b). It is remarkable that different channel heights or the corresponding aspect ratio of the channel’s rectangular cross-section cause a notable impact on the mixing efficiency (Figure 3c). One of the combinations of these variables (channel width = 100 µm, height = 30 µm, curvature θ = 45°, and R = 600 µm) seemed to produce a very promising mixing ratio, and we used these parameters for experimental measurement.

### 3.2. Numerical Simulation and Experimental Results

The mixing relied on the creation of interfaces between different reagents and the transformation of them through effects such as split and stretch. To demonstrate the increasing number of liquid/liquid interfaces along the SAR operations, and the stretching of interfaces, we traced the flow trajectories through CFD simulation. Two input solutions are labeled red and blue, and some input positions are traced through the whole device, labeled as red and blue dots. The initial interface between two liquids was also traced through three sets of dots colored red, green, and blue (Figure 4a–c).

Initially, there is only one interface at the inlet, then the number of interfaces doubled after the first mixing cycle and doubled in each succeeding cycle. Besides the exponentially increased number of interfaces, since these interfaces are stretched, they will also additionally facilitate the mixing through more effective diffusion.

Such an accelerated mixing process can be quantitatively measured through imaging-based analysis of cross-sections of the PDMS-made devices. Since diffusion rate is a major factor in the final mixing performance, we first employed glycerin solution (*v*/*v* 65%) as the working fluid. With the viscous glycerin solution and slower diffusion of large molecular protein IgG, the interfaces can be stably preserved and hence allow for easier observation. The confocal images (Figure 4d) clearly show the distribution of fluids and the formation and transformation of diffusion interfaces along the mixing cycles. The imaged patterns and interfaces are highly consistent with the numerical analyses at corresponding positions. In addition, although it was challenging to fabricate the perfect rectangular channels using soft lithography, minor defects, or deformation of PDMS did not affect the general agreement between the experimental and the simulated results.

We next use water as the working fluid and calculated the mixing ratio of two fluids at the end of each mixing cycle using the confocal images (Figure 4e). The calculated mixing ratio achieved 91% after four cycles at 2 mm/s velocity, which creates a satisfactory mixing efficacy for most applications within a 10 mm device. This accordant Péclet number is 8.7 × 10^4^. Moreover, this mixing performance could be further enhanced with an extra two or four cycles.

Since the mixing performance is also determined by the flow speed, we hence measured the mixing ratio at different flow velocities (Figure 5). As expected, the fast flow will lead to less efficient mixing, as the diffusion time has been shortened in the mixer. When the flow velocity is set at 1 mm/s (0.024 µL/s), which is a typical value in biochemical assays on-chip, the mixing ratio is 94% for goat anti-rabbit IgG in water for a mixer with four SAR units. The total volume of this four-unit SAR micromixer is about 0.2 µL, and the length of the layout is less than 10 mm. Such a compact design and small dead volume make this design suitable for integration with various functional modules in microfluidic devices.

Both linear velocity and volumetric flow rate are important to the mixing. According to the simulation results, the inlet linear velocity of the fluid does not impact the contours of fluidic distributions. Fast flow velocity induced less contacting time between two fluids within the mixer, thus limiting their diffusion among interfaces. The volumetric flow rate is determined by both linear velocity and the area of the inlet cross-section. The channel size, namely their width or height, also limits the effects of diffusions for the same interface patterns. It is recommended to introduce extra mixing units for higher volume rate applications.

### 3.3. Mixing Mechanism Dissected Using Streamline Particle Trajectory Tracing

We further analyzed the details of fluidic transformation during SAR; we summarized the mixing process to three major effects: combination, rotation, and stretching of interfaces (Figure 6). For ease of tracing the flow, we use different colors to label specific parts of the liquid in the channel, hence the re-distribution in one SAR cycle can be stepwise visualized. ‘Combination’ is the effect that combines two fluids originally located in opposite corners of the cross-section and forms an interface which is the direct consequence and intention of SAR design. ‘Rotation’, however, arranges two other parts of the fluid zones and streams into the opposite corner positions, which can then be further transformed through combination in the next cycle. The ‘Stretching’ effect exerts on the interfaces generated by the combination effect. Those newly formed interfaces will be stretched along the diagonal directions and split into two and even more interfaces at the end of the following SAR cycles.

All these effects jointly enhance the mixing significantly. Two fluids separately introduced at the beginning position of the first SAR mixing unit can be well mixed after four mixing cycles. Different parts of the fluid in the inlet will eventually mix well at some point before the end of the fourth cycle. We hence analyzed the flow pattern of each cross-section and divided the input cross-section into four mixing zones (Figure 7). As the result shows, the mixing zones transform their locations using highly similar ways in sequence. There are two major characteristics of our SAR mixer. One is that certain parts of the liquid will mix faster than other parts. For example, the liquid in ‘Mixing Zone 1’ will be greatly stretched to form a large interface along the diagonal plane after one cycle and greatly reduce the diffusion distance, and then further split into two, four, and eight interfaces in the following SAR cycles. However, the liquid in ‘Mixing Zone 4’ is less efficiently mixed until brought to the diagonal position in the fourth SAR cycle. In addition, principally, this mixer design could achieve near complete mixing using four mixing cycles. Therefore, more thorough mixing could be achieved by introducing an extra two or four mixing cycles.

### 3.4. Radial Pressure Gradients in Curvature Channels Induced Advection in Overlapping Channels

The mixing process and three effects discussed above are driven by two underlying mechanisms: the advection on the boundary of overlapping channels; and the reallocation of fluids in the split and recombinant transformation. Therefore, we explored the nature of advection by inspecting the pressure distribution in typical planes along the mixer. The curved channels are important components to generate advection that greatly improves mixing efficiency. In the CFD simulation analysis, the input, output, and junction region planes all had uniform pressure as well as the planes along the radial direction. While for the planes in parallel to the inlet/outlet, which are more interesting, there exist transverse pressure gradients (Figure 8a). The pressure distribution of these planes differs for those cross-sections close to the inlet or outlet. In those regions where two layers of channels are merged, the pressure imbalance will effectively introduce advection between two layers (Figure 8b). Such advection, along with the fluid switch between layers, can bring the distal flow streams together. In each unit, such an effect happens in both separation and recombination parts. This pressure distribution of overlapped channels causes advection between them, and the direction is consistent with separation and combination parts. This advection phenomenon combined with SAR brings three effects discussed above.

The Dean effect is reported to generate mixing effects in spiral microchannels [31]. However, the Dean number in our curved portions of the channels is quite small (0.028). Besides, we do not detect an obvious Dean effect in our analysis. As in Figure 6, fluid distributions are not altered in the “Layer Switch” stage during which they go through two sequential curved channels.

## 4. Conclusions

In this work, we developed an easily fabricated and highly efficient 3D micromixer based on a novel split-and-recombination (SAR) structure. Numerical simulation was performed to optimize the structural design and to understand the mechanisms of mixing. The high-resolution cross-section experimental result, acquired through confocal microscopy, well matches numerical simulations, clearly demonstrating the growing number and dispersed distributions of interfaces. Our SAR-based micromixer shows extremely high mixing efficiency in both fluid simulation and experiment results. The micromixer enables nearly complete mixing for fluid with a Péclet number of 8.7 × 10^4^, using only four repeated cycles within a device length of 10 mm. Within this design, mixing is facilitated by three major fluid transformation effects including rotation, combination, and stretching, to newly create, expand, and split the diffusion interfaces. The quantity of these interfaces is exponentially increased with the number of periodical SAR units. We further endeavored to investigate the mixing mechanism and found the pressure gradient of curvature channel-induced advection between overlapped channels. This simple but effective micromixer, with compact size and easiness of fabrication, can be seamlessly integrated into most microfluidic devices when fast and thorough liquid mixing is needed.

## Figures and Tables

**Figure 1 micromachines-13-01720-f001:**
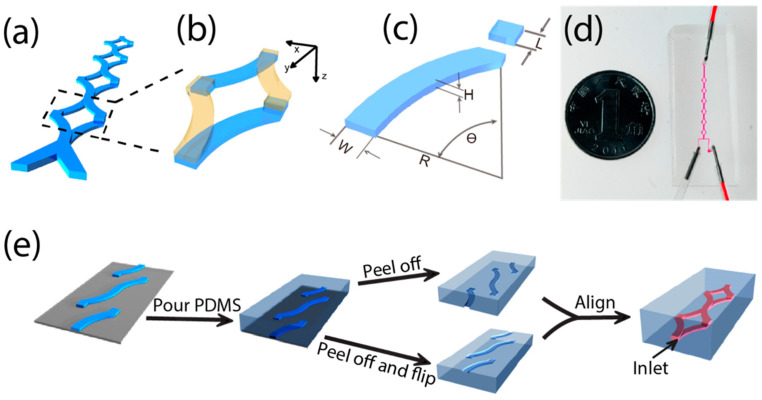
Schematic and fabrication of the mixer. (**a**) The 3D structure diagram of a mixer with two inlets and four SAR mixing cycles. (**b**) One cycle of the mixer that is constructed with two layers of microchannels. (**c**) Key structure parameters for the single unit of a mixer. (**d**) Image of a mixer with the channel filled with colorant. (**e**) Fabrication process of the mixer device.

**Figure 2 micromachines-13-01720-f002:**
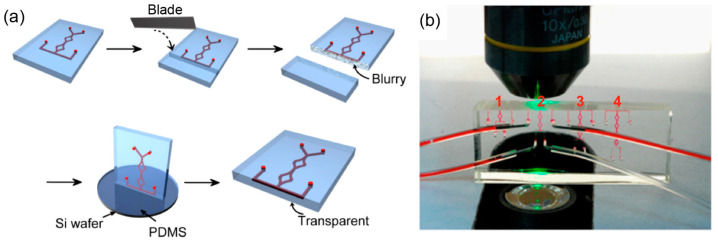
Method for high-resolution imaging of the fluid mixing distribution on cross-sections of the outlet of serial mixing cycles with high resolution. (**a**) Procedures to make the side surface of the chip optically transparent for observing the mixer outlet cross-sections. The extra marginal PDMS near the outlet was cut and removed. The exposed blurry surface was further attached to a thin layer of uncured PDMS. After the crosslinking, the chip was peeled off from the wafer with a transparent outlet surface exposed. (**b**) Setup for imaging the cross-section of the chip’s channel outlets of the first four cycle numbers (red numbers) with confocal microscopy.

**Figure 3 micromachines-13-01720-f003:**
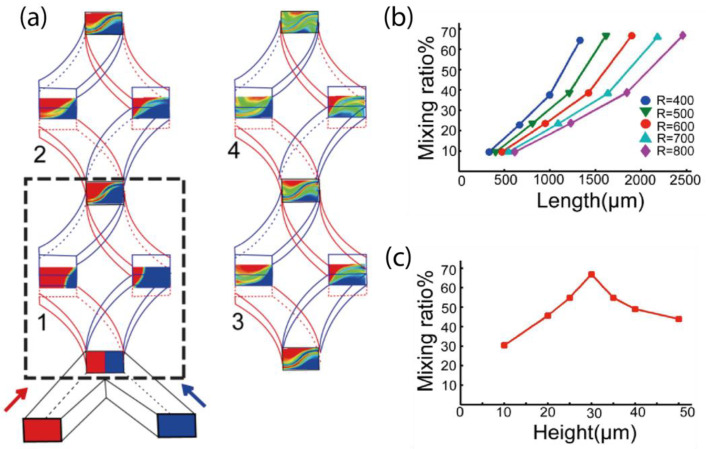
Optimization of mixer structure parameters and mixing ratio. (**a**) Mixing process during the first four cycles of the mixer. (**b**) For each radius R (in µm), the mixing ratio of the first four cycles is calculated. The length represents the distance from the inlet to the outlet for each of the first four cycles. (**c**) The mixing ratio with different channel height when W = 100 μm, θ = 45°, R = 600 μm, L = 100 μm.

**Figure 4 micromachines-13-01720-f004:**
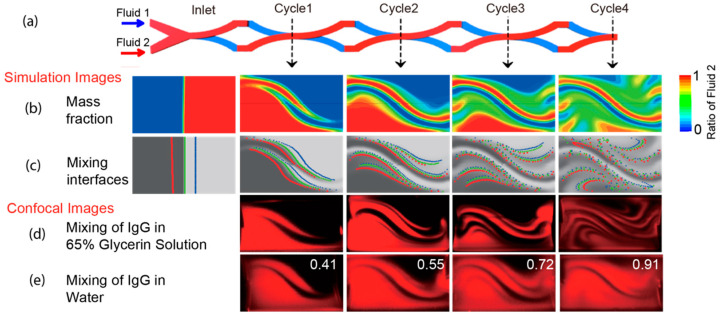
Simulation analysis and confocal images of the mixer in cross-sections of the first four cycles. (**a**) The schematic of input fluids and four outlet planes from 1–4 mixing cycles. (**b**) Simulated mass fraction distributions (calculated as the ratio of Fluid 2) of each outlet plane. (**c**) The growing number and length of mixing interfaces, which are traced and visualized using three sets of dots (colored in red, green, and blue) from the boundaries of two fluids at the inlet plane. (**d**) Confocal images of four outlet planes for mixing of IgG in 65% glycerin solutions. The interfaces are clearly visualized due to the slow protein diffusion in glycerin. (**e**) The confocal images of four planes for mixing IgG in water with the calculated mixing index labeled.

**Figure 5 micromachines-13-01720-f005:**
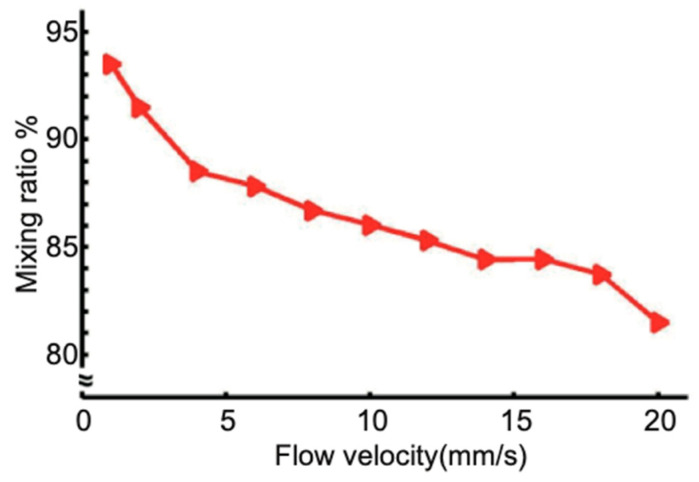
The relationship between mixing ratio after 4 cycles and flow velocity.

**Figure 6 micromachines-13-01720-f006:**
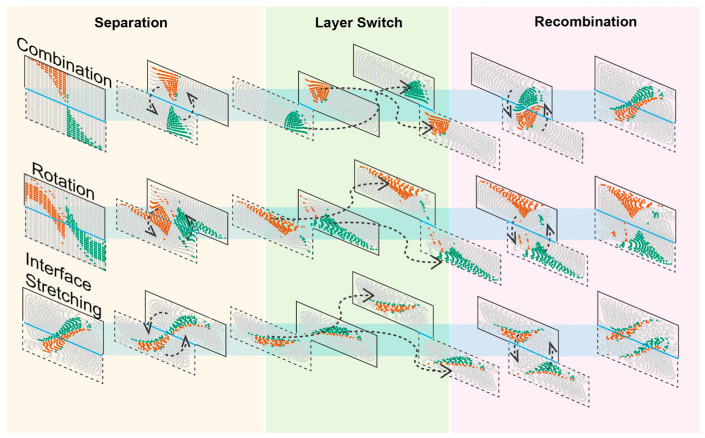
Three mixing effects in the mixing unit. Specific sections or zones from Fluid 1 and Fluid 2 inputs are colored brown and green, respectively. These zones are transformed in separation, layer switch, and recombination stages in the mixing unit/cycle.

**Figure 7 micromachines-13-01720-f007:**
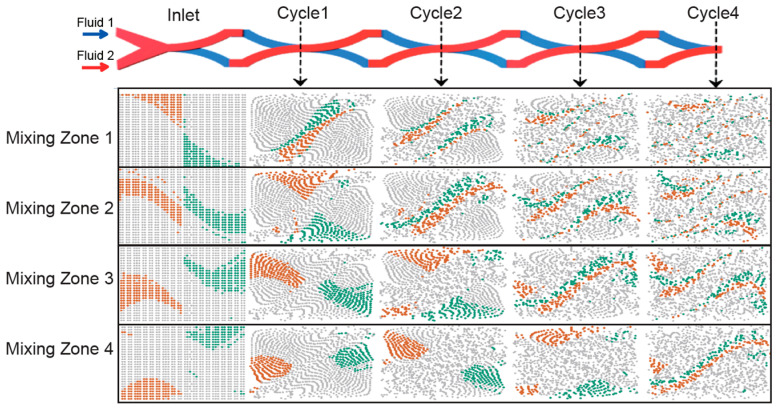
The inlet plane is composed of four mixing zone groups, which are fully mixed in different cycles sequentially. For each mixing zone group, its Fluid 1 and Fluid 2 sections are colored brown and green, respectively. The transformation and reallocation of fluid distributions in the first four mixing cycles are visualized accordingly.

**Figure 8 micromachines-13-01720-f008:**
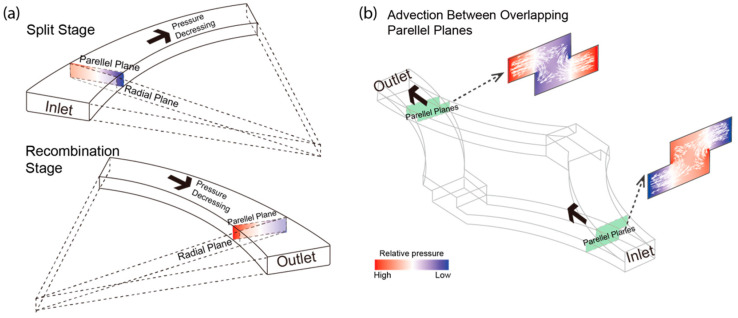
Mechanism of advection between overlapped curvature channels. (**a**) Schematic of pressure distribution in the inlet, outlet, radial plane, and parallel planes from the Split (upper) and Recombination (lower) stage of the mixing process. The pressure gradients in the parallel planes are different depending on the mixing stage. (**b**) The pressure gradient of parallel planes from overlapped channels induces advection between them, and the direction is consistent with separation and combination parts.

## Data Availability

Not applicable.

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
