# Peer review of "Design and High-Resolution Analysis of an Efficient Periodic Split-and-Recombination Microfluidic Mixer"

_micromachines, 2022, doi:10.3390/mi13101720_

Round 1
Reviewer 1 Report
Dear Authors,
I have found the submitted manuscript well presented and of interest to readers of this journal. The novelty of the work may not be outstanding, but I agree with the authors that the simplicity in design and fabrication is noteworthy and is of interest to the microfluidic mixer and flow dynamics community. Also, agree that the characterization technique with the side-wall formation to image the cross-section along the lengths is an impactful art that is being shared. The results and conclusions are sufficiently deep in terms of explaining fundamental transport phenomena ( namely ; combination; rotation; stretching of interfaces)
I found some minor typos that could be corrected. Please see attached.
Regards

Author Response
Reviewer 1: I have found the submitted manuscript well presented and of interest to readers of this journal. The novelty of the work may not be outstanding, but I agree with the authors that the simplicity in design and fabrication is noteworthy and is of interest to the microfluidic mixer and flow dynamics community. Also, agree that the characterization technique with the side-wall formation to image the cross-section along the lengths is an impactful art that is being shared. The results and conclusions are sufficiently deep in terms of explaining fundamental transport phenomena ( namely ; combination; rotation; stretching of interfaces)
I found some minor typos that could be corrected. Please see attached.
Response: Thank you for the appreciation of our work. We have corrected all the minor typos and other concerns mentioned in the revised manuscript.
Reviewer 2 Report
This is a review of the manuscript numbered micromachines-1951937 titled “Design and high-resolution analysis of an efficient periodic split-and-recombination microfluidic mixer.” Authors describe a microfluidic mixer that relies on periodic splitting and recombination of the flows in vertical and horizontal planes. Mixing fluids on a microscale could be complicated. In the last two decade there were quite a few designs and approaches developed, including splitting in x-y plane, flow redirection in z-dimension, and use of curved channels. It appears that the design of the proposed here mixer combines all three of the approaches with good results. Manuscript is interesting and is reasonably well written.
Recommendation: publish with minor revisions.
Comments:
1. Line 150: 20 mm/s. Could authors also include volumetric flow rate. In general, since there were several channel dimensions tested it would be nice to have not only linear velocity but also corresponding volumetric flow rate.
2. Section 2.2. Please add channel dimension that were fabricated.
3. Section 2.3, Line 118. 2 mm/s: what volumetric flow rate does that correspond to?
4. Could you please add a brief discussion to Section 3 (maybe around Fig 4 or 5) of what is important for the mixing: linear velocity of the fluid or volumetric flow.
5. Some important references are missing. Please add them to the manuscript and potentially discuss how your approach is complimentary / different.
M. A. Stremler et al “Designing for chaos: applications of chaotic advection at the microscale” Phil. Trans. R. Soc. Lond.A (2004) 362, 1019–1036
Beebe et al “Passive mixing in microchannels: Fabrication and flow experiments” Mec. Ind. (2001) 2, 343–348
Liu et al “Passive Mixing in a Three-Dimensional Serpentine Microchannel” J. Of Microelectromechanical Systems, Vol. 9, No. 2, June 2000
6. For the curved portions of the channels what are the Dean numbers? How strong do you think curved channels contribute to mixing and fluid interface stretching?
Sudarsan et al “Fluid mixing in planar spiral microchannels” Lab Chip, 2006, 6, 74–82
Author Response
- Line 150: 20 mm/s. Could authors also include volumetric flow rate. In general, since there were several channel dimensions tested it would be nice to have not only linear velocity but also corresponding volumetric flow rate.
Response: Thank you for the suggestion. We have added the volumetric flow rate (see the manuscript, Line 127, 158, 240).
- Section 2.2. Please add channel dimension that were fabricated.
Response: Thank you for the suggestion. We have added the description of the channel dimensions that were fabricated (see the manuscript, Line 112-116).
- Section 2.3, Line 118. 2 mm/s: what volumetric flow rate does that correspond to?
Response: Thank you for this question. The 20 mm/s corresponds to 0.12 µL/s in the numerical simulation mixer models (Line 127).
- Could you please add a brief discussion to Section 3 (maybe around Fig 4 or 5) of what is important for the mixing: linear velocity of the fluid or volumetric flow.
Response: Thank you for the suggestion. We have added the discussion before the section 3.3, between Line 245 and Line 252.
- Some important references are missing. Please add them to the manuscript and potentially discuss how your approach is complimentary / different.
A. Stremler et al “Designing for chaos: applications of chaotic advection at the microscale” Phil. Trans. R. Soc. Lond.A (2004) 362, 1019–1036
Beebe et al “Passive mixing in microchannels: Fabrication and flow experiments” Mec. Ind. (2001) 2, 343–348
Liu et al “Passive Mixing in a Three-Dimensional Serpentine Microchannel” J. Of Microelectromechanical Systems, Vol. 9, No. 2, June 2000
Response: Thank you for the suggestion. We have added these three references into the introduction part as No. 16, 29, and 28, respectively, along with the discussion regarding the design and analysis.
- For the curved portions of the channels what are the Dean numbers? How strong do you think curved channels contribute to mixing and fluid interface stretching?
Sudarsan et al “Fluid mixing in planar spiral microchannels” Lab Chip, 2006, 6, 74–82
Response: Thank you for the questions. We have added the discussion on the Dean Effect and the Dean number in our curved portions of the channels (see manuscript Line 310 to Line 314). We have also added the suggested reference as No. 31. In brief, we do not find notable mixing patterns caused by Dean Effect due to the low Dean number (0.028).